# Intraoperative calculus or hemorrhage in transurethral seminal vesiculoscopy as a risk factor for recurrent hemospermia

**Cheng-En Mei**[1,2], **Ju-Chuan Hu**[2], **Jian-Ri Li**[2], **Kun-Yuan Chiu**[2], **Shian-Shiang Wang**[2], **Chuan-Shu Chen**[2]*

**1** Division of Traumatology, Department of Emergency, Taichung Veterans General Hospital, Taichung, Taiwan, **2** Division of Urology, Department of Surgery, Taichung Veterans General Hospital, Taichung, Taiwan

* r2060d@gmail.com

**Data Availability Statement:** All relevant data are within the paper and its Supporting information files.

## Abstract

We have summarized our experience regarding transurethral seminal vesiculoscopy (TUSV) and analyzed both its recurrence status and the risk factors for recurrence. From January 2010 to December 2020, 48 patients with intractable hemospermia received successful TUSV at Taichung Invalids General Hospital. Upon analysis of the intraoperative findings, the five-year disease-free Survival rates (DFS) were 74.1% in the no calculus group compared to 37.1% in the calculus group with a significant difference (log-rank $p = 0.015$), 75.0% in the no hemorrhage or no blood clot group compared to 43.2% in the hemorrhage or blood clot group with significant difference (log-rank $p = 0.032$). Univariate analysis showed intraoperative calculus ($p = 0.040$; HR: 2.94, 95% CI: 1.05–8.21) to be significantly associated with recurrence ($p < 0.05$). Patients with intractable hemospermia who were diagnosed with stones or blood clots found during TUSV experienced a higher rate of hemospermia recurrence.

## Introduction

Hemospermia, hematospermia or haematospermia are all defined as blood appearance in the semen [1]. The symptom usually resolves spontaneously in most cases. The mechanisms surrounding the occurrence of hemospermia can be inflammation, infection, lithiasis, cyst formation, obstruction, tumor, vascular related, trauma, iatrogenic related and systemic origin [2, 3]. Recommended imaging studies include transrectal ultrasound (TRUS), computed tomography (CT), and magnetic resonance imaging (MRI) [2]. TRUS is less expensive and less diagnostic than the other tests, but is more commonly used as a diagnostic tool [4, 5].

Wang et al. first described intractable hemospermia as a condition which has persisted for more than 4 months despite medical treatment [6]. Recent studies have defined intractable hemospermia as a condition which has persisted for at least 3 months, with conservative therapy providing to be unsuccessful [7–11]. Patients often experienced severe anxiety due to the condition, which in turn affected their psychological health and quality of sexual life.

**Funding:** The author(s) received no specific funding for this work.

**Competing interests:** The authors have declared that no competing interests exist.

Through advancements in endoscopy technology, in vivo transurethral seminal vesiculo-scopy (TUSV) was first introduced in 2002 [12]. Subsequently, TUSV has been used as a diagnostic and treatment procedure for recurrent hemospermia, persistent hemospermia and intractable hemospermia [9, 11, 13–16]. TUSV also has a higher diagnostic rate than TRUS [4].

The recurrence rate after TUSV treatment has been mentioned to be in a range from 3.4% to 11.76% [9, 11, 13–16]. To the best of our knowledge, there are currently no studies investigating recurrence after TUSV. In this study, we sorted through 48 successful cases and analyzed their recurrence status and risk factors for recurrence.

## Materials and methods

### Patients

This study enrolled patients who had been diagnosed with intractable hemospermia and received successful TUSV treatment during the period from January 2010 to December 2020 in Taichung Veterans General Hospital, Taiwan, Republic of China. The diagnosis of hemospermia relied on photos of semen which patients took after sexual activity. The enrolled patients took a serum prostate-specific antigen (PSA), coagulation tests and TRUS.

For treatment of hemospermia, the empiric antibiotic ciprofloxacin, 400 mg every 12 hours per os (PO), was administered for at least 2 weeks. For patients with intractable hemospermia who were willing to undergo surgery in order to relieve symptoms, we provided TUSV.

This is a retrospective study. All patient records and data were fully anonymized and de-identified prior to analysis. Therefore no informed consent was deemed necessary. The study protocol conformed to the ethical guidelines of the 1975 Declaration of Helsinki, and was granted by the ethics committee of Taichung Veterans General Hospital, Taiwan, Republic of China. The institutional review board number was CE22060A.

### Surgical techniques

Each patient received either general anesthesia or spinal anesthesia and was placed in the lithotomy position. A semi-rigid ureteroscope (6/7.5-Fr; Olympus, Tokyo, Japan) was inserted into the urethra and introduced to the verumontanum. The bilateral seminal vesicles were then entered through the ejaculation duct or fenestrated from the utricle. Intraoperative manifestations, such as calculus, hemorrhage and mucosal lesion, were recorded. Seminal vesicle fluid was then collected for a culture exam. Calculus removal or a biopsy would then be performed depending on the intraoperative manifestations. After the procedure, 10ml of normal saline with an 80mg gentamycin irrigation into the seminal vesicle would then be done [9].

### Statistical analysis

Numeric variables are expressed as medians (interquartile ranges) and subsequently compared using the Chi-square test, with significance set at $p < 0.05$. Continuous variables are presented as medians (ranges) and compared using the Mann-Whitney U test, with significance also set at $p < 0.05$. Rates of disease-free survival (DFS) up until January 2022 were calculated using the Kaplan-Meier life table method and compared across groups using the log-rank test. Differences with $p$ values $< 0.05$ were regarded as statistically significant. Univariate and multivariate analysis utilizing Cox proportional hazard ratios were derived for the outcomes of interest. All $p$ values $< 0.05$ were considered significant in univariate analysis and thus included in multivariate analyses.

## Result

Forty-eight (48) patients diagnosed with intractable hemospermia who underwent successful TUSV were studied. All patients achieved remission. Sixteen (16) patients (33.3%) underwent recurrence up until January 2022. The characteristics between recurrence and non- recurrence are shown in Table 1. In TURS findings, twenty two (22) non-recurrent patients (68.8%) experienced overall calculus, while thirteen (13) recurrent patients (86.7%) had overall calculus. Eight (8) non-recurrent patients (25%) had seminal vesicle (SV) or ejaculation duct calculus, while two (2) recurrent patients (13.3%) had SV or ejaculation duct calculus. The median follow-up time was 39.4 months in the non- recurrent group and 49.8 months in the recurrent group. Complications such as epididymitis occurred in only two (2) cases in the non-recurrent group. The median follow-up period was 40.1 months (range, 0.95–134.5 months).

### Intraoperative findings

Intraoperative findings of TUSV between the recurrence and non-recurrence groups are shown in Table 2. Overall calculus was found in the utricle or SV in ten (10) non-recurrent patients (31.3%) and nine (9) recurrent patients (56.3%). Hemorrhage was found in the utricle or SV in eleven (11) non-recurrent patients (34.4%) and ten (10) recurrent patients (62.5%). Mucosal lesion was found in the utricle, SV, urethra or veru montenum in four (4) non-recurrent patients (12.5%) and three (3) recurrent patients (18.8%). A biopsy report showed 1 fibroepithelial polyp, 1 amyloidosis, with the remainder being either congestion or inflammation.

**Table 1. Characteristics of patients between recurrence and non-recurrence.**

| Total (n = 48) | Non-recurrence (n = 32) | Recurrence (n = 16) | p value |
|---|---|---|---|
| **Age** | 54.5 (48.3–63.8) | 53.5 (38.3–58.8) | 0.330 |
| **Duration (months)** | 12.0 (4.0–23.3) | 13.0 (6.0–24.0) | 0.676 |
| **Diabetes mellitus** | 2 (6.3%) | 1 (6.3%) | 1.000 |
| **Hypertension** | 6 (18.8%) | 4 (25.0%) | 0.712 |
| **Previous urinary tract infection** | 2 (6.3%) | 1 (6.3%) | 1.000 |
| **Urolithiasis history** | 5 (15.6%) | 1 (6.3%) | 0.648 |
| **Erectile dysfunction** | 4 (12.5%) | 1 (6.3%) | 0.652 |
| **Sexually transmitted disease** | 1 (3.1%) | 0 (0.0%) | 1.000 |
| **Anti-platelet agent** | 2 (6.3%) | 3 (20.0%) | 0.309 |
| **PSA** | 1.1 (0.6–1.4) | 1.0 (0.7–1.7) | 0.803 |
| **Digital rectal examination** | | | |
| Elastic consistency | 32 (100%) | 16 (100%) | - - |
| Hard consistency | 0 (0%) | 0 (0%) | - - |
| **TRUS findings** | | | |
| Overall calcification | 22 (68.8%) | 13 (86.7%) | 0.288 |
| Prostate calculus | 19 (59.4%) | 13 (86.7%) | 0.094 |
| SV or ejaculation duct calculus | 8 (25.0%) | 2 (13.3%) | 0.465 |
| **Post operative complication** | | | |
| Epididymitis | 2 (6.3%) | 0 (0.0%) | 0.546 |
| Perineal pain | 0 (0.0%) | 0 (0.0%) | - - |
| **Follow-up period (months)** | 39.4 (1.6–66.1) | 49.8 (20.9–96.7) | 0.120 |
| **Time to remission (weeks)** | 4.0 (3.0–4.0) | 4.0 (4.0–7.75) | 0.097 |

Chi-square test or Mann-Whitney U test, Median (IQR). $^{*}p<0.05$, $^{**}p<0.01$

Numeric variables expressed as medians (interquartile ranges)

Seminal vesicle (SV).

**Table 2.. Intraoperative findings of TUSV between recurrence and non-recurrence.**

| Total (n = 48) | Non-recurrence (n = 32) | Recurrence (n = 16) | *p* value |
|---|---|---|---|
| **Intraoperative findings** | | | |
| **Overall calculus** | 10 (31.3%) | 9 (56.3%) | 0.175 |
| **Utricle** | 10 (31.3%) | 7 (43.8%) | 0.594 |
| **Left SV** | 3 (10.0%) | 4 (26.7%) | 0.199 |
| **Right SV** | 2 (6.9%) | 1 (6.3%) | 1.000 |
| **Hemorrhage or blood clot** | 11 (34.4%) | 10 (62.5%) | 0.123 |
| **Utricle** | 2 (6.3%) | 0 (0.0%) | 0.546 |
| **Left SV** | 6 (20.7%) | 7 (46.7%) | 0.092 |
| **Right SV** | 4 (14.3%) | 5 (31.3%) | 0.250 |
| **Mucosal lesion** | 4 (12.5%) | 3 (18.8%) | 0.672 |

Chi-square test. $^{*}p<0.05$, $^{**}p<0.01$

Seminal vesicle (SV).

### Disease free survival

Regarding the TRUS findings, five-year disease-free survival (DFS) rates were 75.0% in the no calculus group compared to 59.0% in the overall calculus group, with the Kaplan–Meier survival curves represented in Fig 1 (log-rank $p = 0.273$). The five-year DFS rates were 54.5% in the no SV or ejaculation duct calculus group compared to 100.0% in the SV or ejaculation duct calculus group, with the Kaplan–Meier survival curves represented in Fig 2 (log-rank $p = 0.227$). In this study, the five-year DFS rates taken from the TRUS findings were insignificantly different.

For the intraoperative findings, the five-year DFS rates were 74.1% in the no calculus group compared to 37.1% in the calculus group, with the Kaplan–Meier survival curves represented in Fig 3 (log-rank $p = 0.015$). The five-year DFS rates were 75.0% in the no hemorrhage or blood clot group compared to 43.2% in the hemorrhage or blood clot group, with the Kaplan–Meier survival curves represented in Fig 4 (log-rank $p = 0.032$). The five-year DFS rates were 73.7% in the negative intraoperative finding group compared to 47.7% in the positive intraoperative finding group, with the Kaplan–Meier survival curves represented in Fig 5 (log-rank $p = 0.093$). In this study, the five-year DFS rates were significantly higher in the intraoperative no calculus group when compared to the calculus group, and also much higher in the no hemorrhage or blood clot group when compared to the hemorrhage or blood clot group. The five-year DFS rate was insignificant but trending upward in intraoperative negative findings when compared to positive findings.

Univariate analysis showed intraoperative calculus ($p = 0.040$; HR: 2.94, 95% CI: 1.05–8.21) to be significantly associated with recurrence, and intraoperative hemorrhage or blood clot ($p = 0.068$; HR: 2.63, 95% CI: 0.93–7.43) to be insignificant but trending upward with recurrence. Multivariate Cox proportional hazard analysis showed intraoperative calculus ($p = 0.051$; HR: 2.80, 95% CI: 1.00–7.90) to be insignificant but trending upward with recurrence.

## Discussion

### Effectiveness of preoperative imaging

TRUS has been mentioned in other studies as having a lower diagnostic yield for hemospermia [4]. In this study, preoperative TRUS provided a low SV or ejaculatory duct stone detection. TUSV provided better diagnostic rates than a TRUS exam alone. This result is consistent with previous research findings. In this study, TRUS results showed no significant difference in

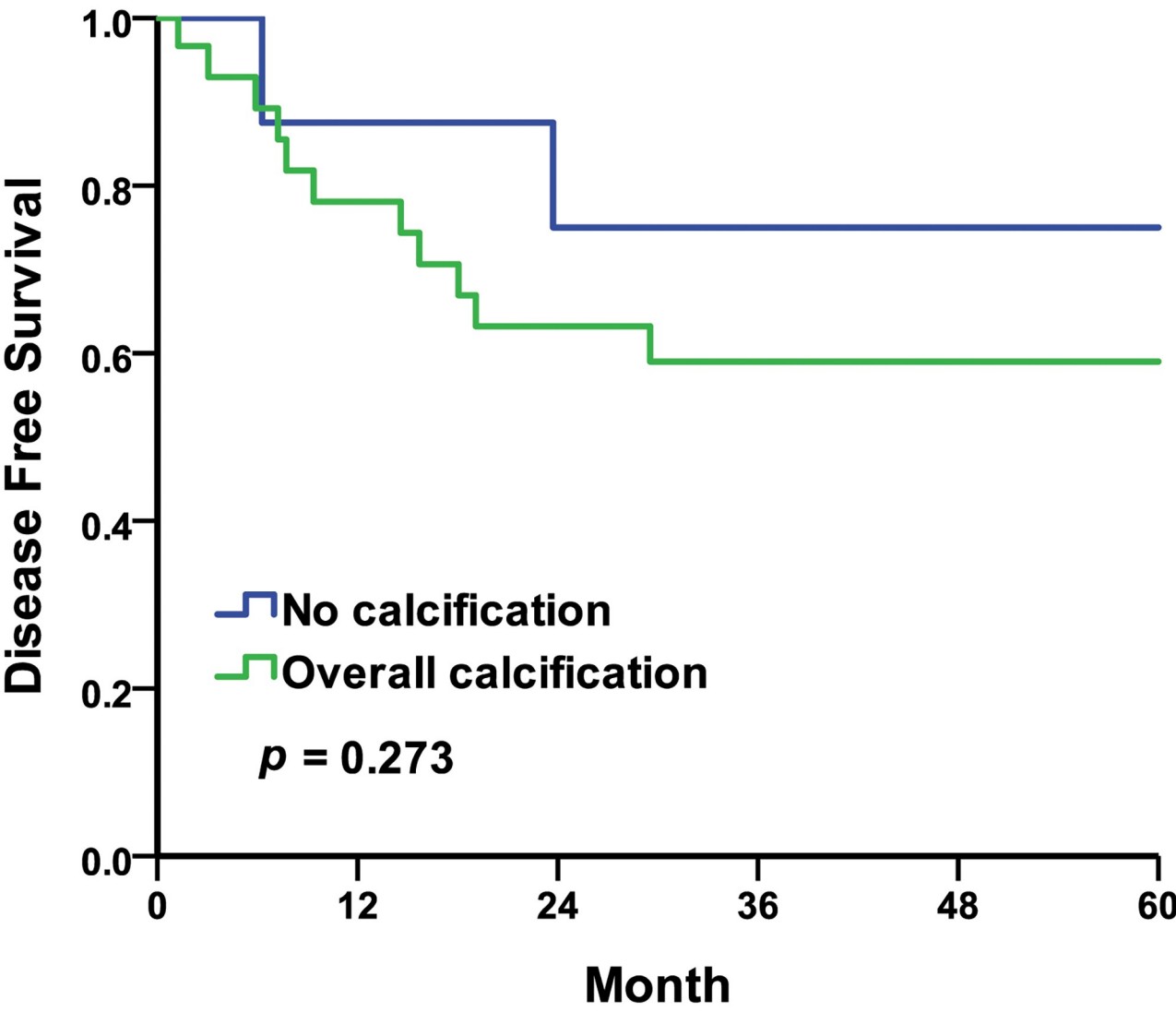

**Fig 1. Kaplan–Meier curves of the disease-free survival rates by TRUS findings of overall calcification.** No calcification compared to overall calcification.

disease-free survival rates. TRUS findings also failed to provide an association with recurrence prediction. Although TRUS is relatively less invasive and carries less cost, the diagnostic significance and benefit of TRUS in patients with persistent hemospermia requires additional follow-up studies.

### Recurrence and hypothesis of hemospermia

The recurrence rate after TUSV treatment is mentioned as being in the range of 3.4% to 11.76% [9, 11, 13–16]. Amongst the 48 patients in this study, 16 had recurrent hemospermia, with the recurrence rate being 33.33%, which was higher than other studies. However, the median follow-up time in our study was 40.1 months, while the follow-up time in other studies was 5–24 months [9, 11, 13–16]. Compared with the previous study performed in our hospital, the follow-up time was 12 months, with 4 of 34 patients experiencing recurrence at a rate of

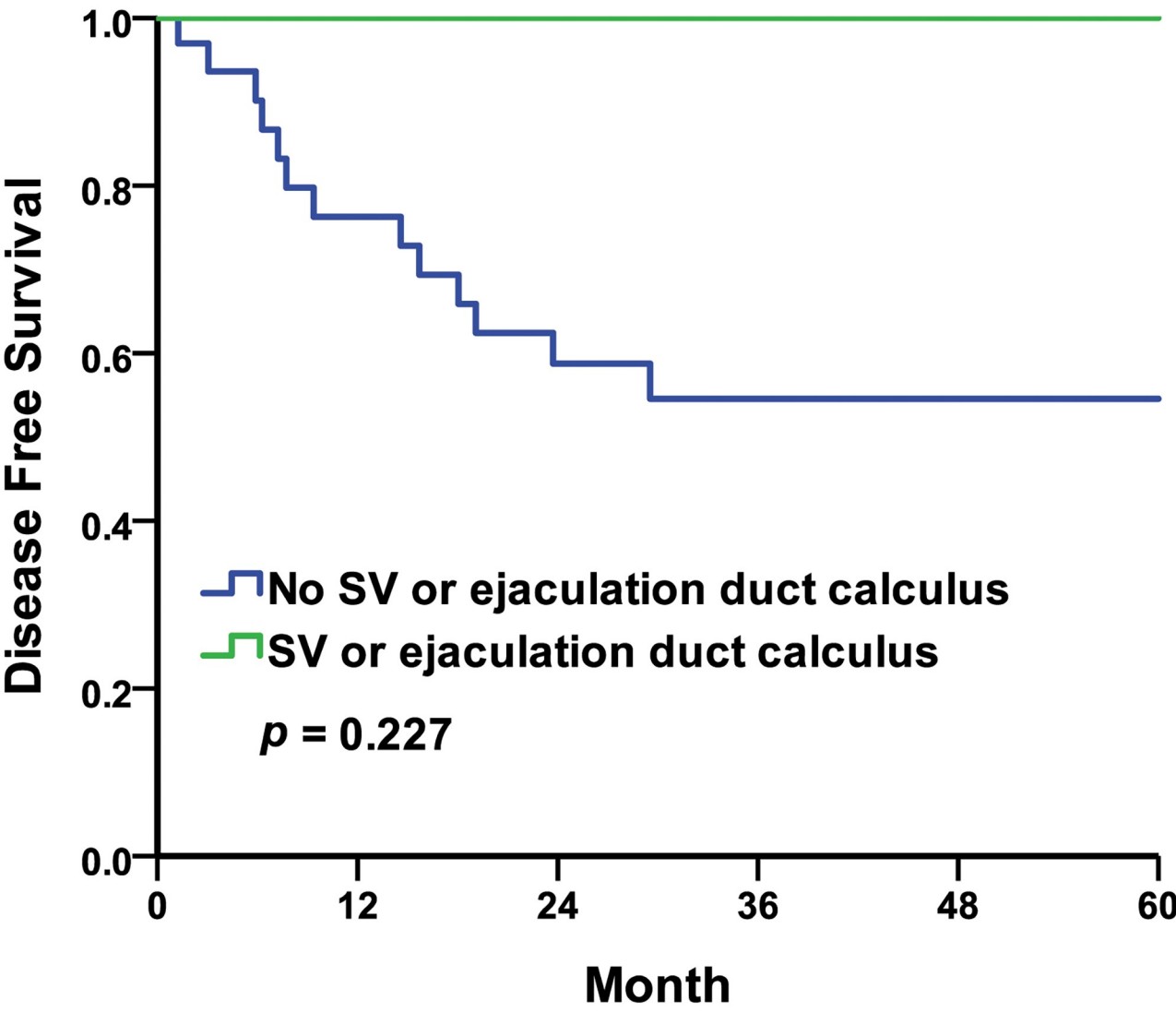

**Fig 2. Kaplan–Meier curves of the disease-free survival rates by TRUS findings of SV or ejaculation duct stone.** No seminal vesicle (SV) or ejaculation duct calculus compared to SV or ejaculation duct calculus.

11.76% [9]. Our patients were more regionalized so follow-up at the same hospital was easier. The high recurrence rate may be related to the long follow-up time.

In our study, recurrence-free survival from intraoperative detected stones, hemorrhages, or blood clot was significantly lower when compared with the undetected group. The current hypothesis is that the occurrence of hemospermia, calculus, strictures and inflammation is a vicious cycle [4, 17]. According to that hypothesis, TUSV could interrupt the vicious cycle in addition to providing a diagnosis. For patients with negative intraoperative findings, we believe that TUSV can block the vicious cycle of stricture and inflammation, thus resulting in symptom relief with low recurrence rates. Inflammation may be worse in patients experiencing stones, hemorrhage, or blood clots as detected by TUSV. Fortunately, TUSV can still stop the vicious cycle and relieve symptoms. Because the inflammation is more severe, the possibility of hemospermia recurrence will also increase.

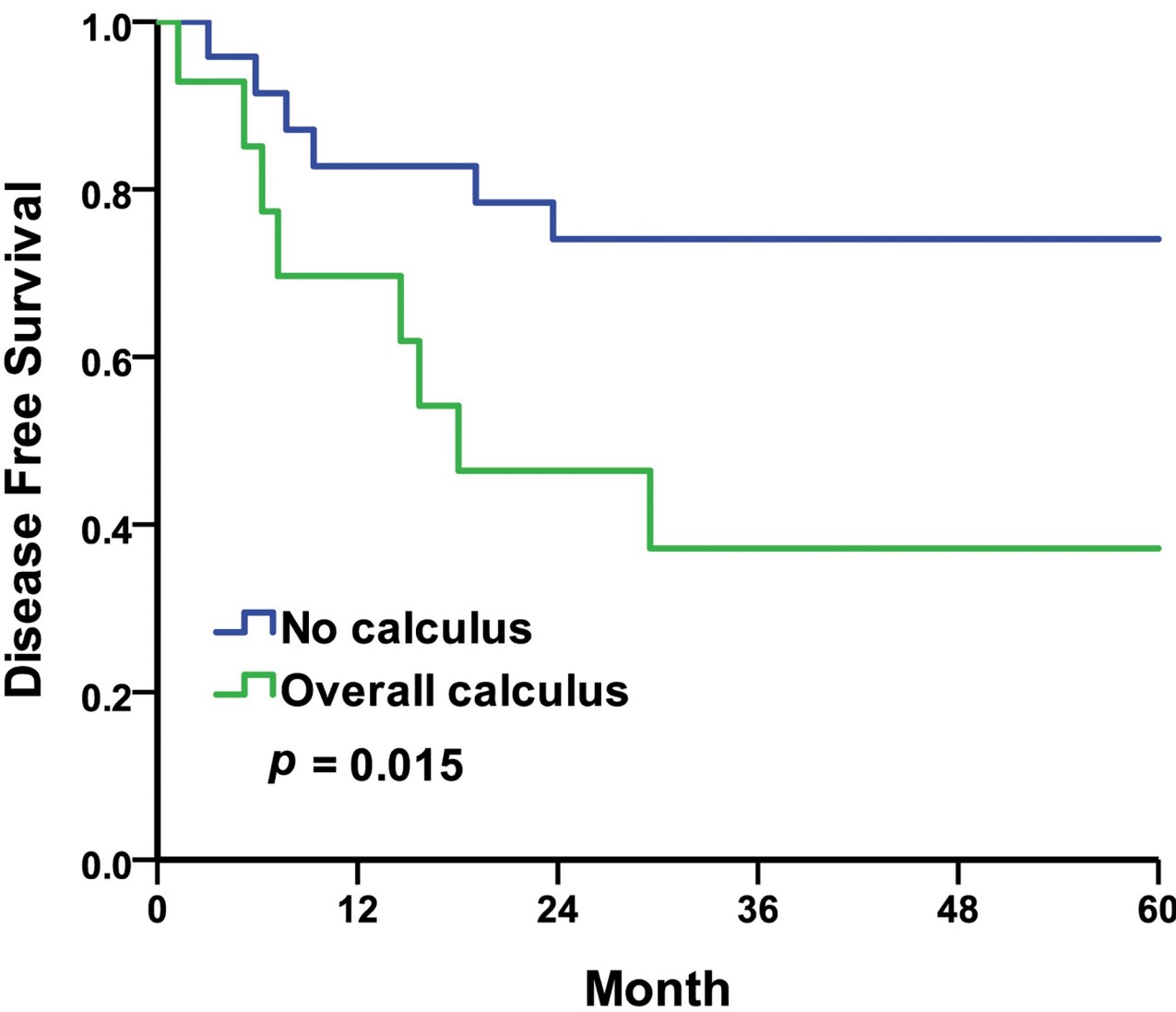

**Fig 3. Kaplan–Meier curves of the disease-free survival rates by intraoperative findings of calculus.** No calculus compared to overall calculus.

### Success rate

The success rate of TUSV treatment, as mentioned in the relevant literature, ranges from 90.9% to 96.53% [9, 11, 15]. In this study fifty one (51) patients underwent TUSV, while 6 of them received second TUSV due to recurrent hemospermia. Among the total 57 TUSV treatments, 54 of which were successfully performed. Such a success rate of 94.7% is comparable with the studies reported in the relevant literature. The 3 patients whose TUSV failed all received MRI exams. It turned out that one of them had a seminal vesicle cyst, while the other 2 patients had small prostate cysts. These structural abnormalities may affect the orientation of the ejaculatory ducts or the location of the seminal vesicles, thereby affecting the success rate of TUSV.

Amongst the 54 successful TUSV cases, only 5 were successfully performed on the unilateral side of SV. Thus, the success rate for both sides of TUSV was 85% (49/57). Upon analysis of these five (5) unilateral TUSV, one (1) patient suffered from recurrence after 71 months,

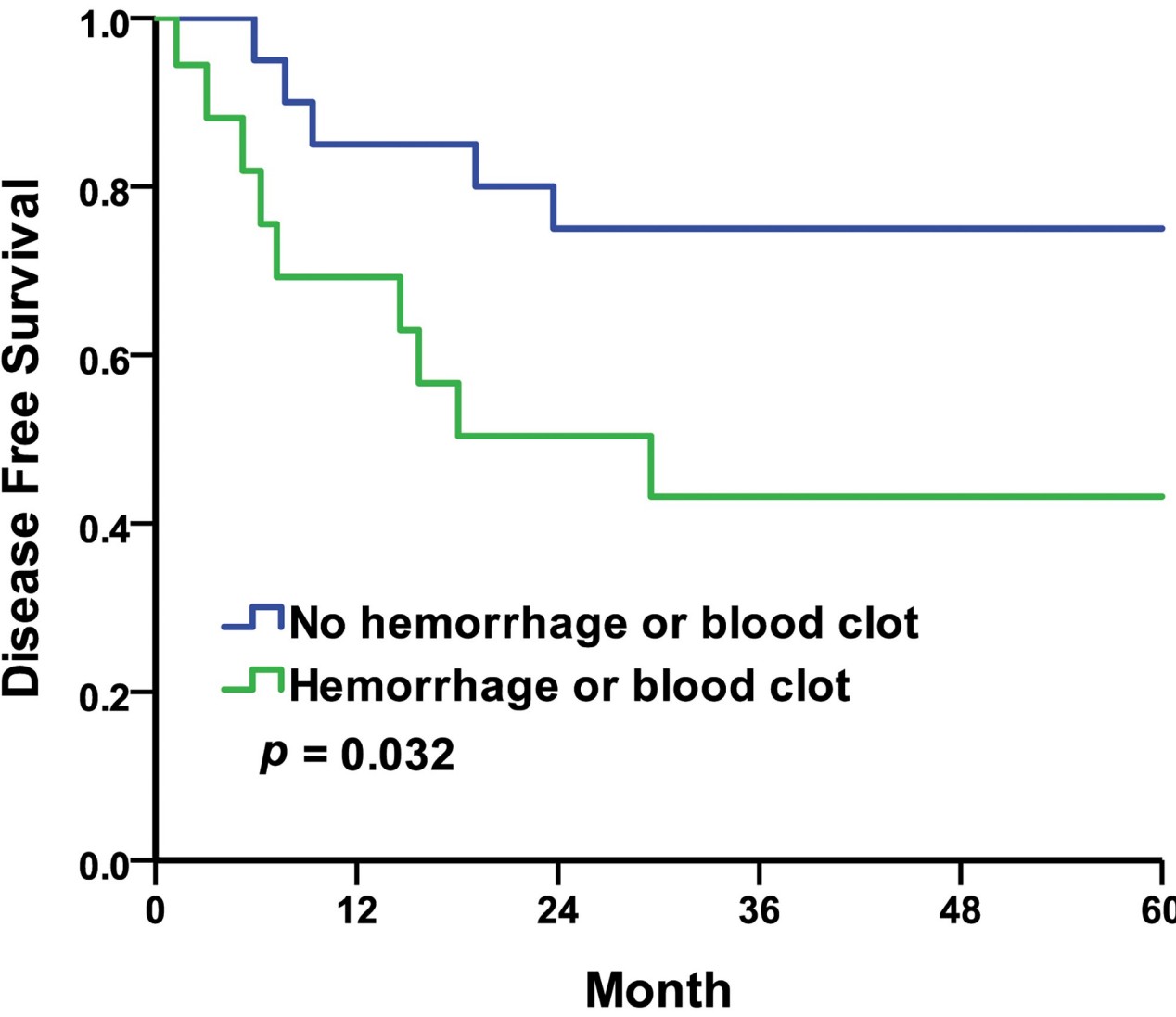

**Fig 4. Kaplan–Meier curves of the disease-free survival rates by intraoperative findings of hemorrhage or blood clot.** No hemorrhage or blood clot compared to hemorrhage or blood clot.

while three (3) patients had symptom remission within 3 or 4 weeks without recurrence. The final one (1) unilateral TUSV underwent a second operation due to recurrence, with bilateral SV successfully entered during this first-time operation. The symptom then subsided in 4 weeks after a second TUSV without recurrence. To the best of our knowledge, there are currently no studies investigating outcomes surrounding unilateral TUSV.

## Safety considerations

Perineal pain, retrograde ejaculation, epididymitis, prolonged hematuria, rectal injury and urinary incontinence are known complications of TUSV [9, 18]. Only 2 patients in this study had postoperative complication of epididymitis. After antibiotic treatment, the infection in these 2 patients was eventually controlled and resolved. Some studies have raised concerns that disruption of normal structures by TUSV could lead to infertility [18]. In

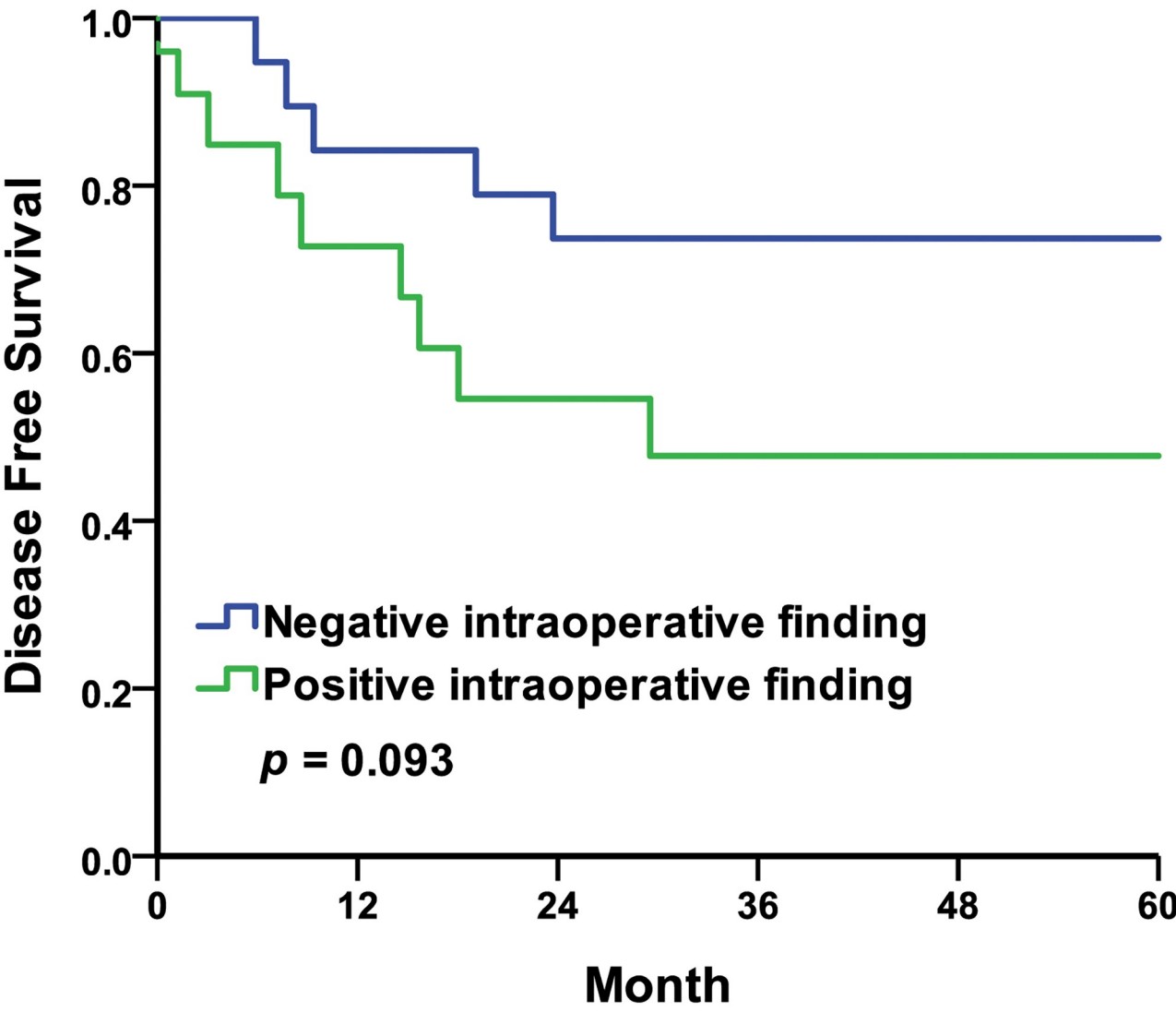

**Fig 5. Kaplan–Meier curves of the disease-free survival rates by intraoperative findings.** Negative intraoperative finding compared to positive intraoperative finding.

recent studies, endoscopic treatments including both TUSV and transurethral ejaculation duct resection (TURED) have shown positive results in the treatment of infertility [16, 19–22]. Real-time TRUS-guided TUSV has also been reported as being a safe procedure that helps avoid rectal injury [23]. Therefore, this study provides strong evidence for the safety of TUSV.

## Limitations and future prospects

The present study is retrospective and with a rather small sample size. Further prospective studies with more patients involved would provide more convincing reference and therefore stronger evidence. In regard to the confounding variables in this study, we only collected data concerning the cases complicated by the use of anti-platelet agent, diabetes mellitus and hypertension. Residual confounding caused by lifestyles, smoking habits and co-medications were

not discussed. Besides the aforementioned limitations, the condition of patients with hemospermia usually improves spontaneously without any aggressive treatment. Furuya et al. reported that the spontaneously resolved rate of hemospermia is as high as 88.9% [24]. It is possible that at the time of analysis, this natural process, i.e. hemospermia being relieved spontaneously, had, to a certain extent, contributed to the excellent remission rate after TUSV as well as complicated the assumed cause-and-effect relationship between TUSV and the remission of hemospermia.

## Conclusion

As analyzed and concluded in this present study, intraoperative findings can help assess the risk of recurrent hemospermia. Specifically speaking, stones or blood clots found during TUSV for patients with intractable hemospermia contribute to a higher rate of recurrent hemospermia.

## Supporting information

**S1 File. Raw data.** File containing data of TUSV follow-up, TRUS findings, Intraoperative findings and recurrence.
(XLSX)

## Acknowledgments

We would like to thank the editors of PLOS ONE and the reviewers for their thoughtful comments that helped us strengthen this article.

## Author Contributions

**Data curation:** Cheng-En Mei, Ju-Chuan Hu, Chuan-Shu Chen.

**Formal analysis:** Cheng-En Mei, Ju-Chuan Hu, Jian-Ri Li.

**Investigation:** Cheng-En Mei, Kun-Yuan Chiu, Shian-Shiang Wang.

**Methodology:** Ju-Chuan Hu, Jian-Ri Li, Kun-Yuan Chiu, Shian-Shiang Wang.

**Project administration:** Chuan-Shu Chen.

**Resources:** Kun-Yuan Chiu, Shian-Shiang Wang, Chuan-Shu Chen.

**Writing – original draft:** Cheng-En Mei.

**Writing – review & editing:** Jian-Ri Li, Chuan-Shu Chen.

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
