## [Decision Letter · Decision Letter 0]

12 May 2022

PONE-D-22-12315Intraoperative calculus or hemorrhage in transurethral seminal vesiculoscopy as a risk factor for recurrent hemospermiaPLOS ONE

Dear Dr. Chuan-Shu Chen,

Thank you for submitting your manuscript to PLOS ONE. After careful consideration, we feel that it has merit but does not fully meet PLOS ONE’s publication criteria as it currently stands. Therefore, we invite you to submit a revised version of the manuscript that addresses the points raised during the review process.

Both reviewers provided constructive suggestions and revision was concluded. Please revise the MS accordingly or discuss the issue that might not be addressed. Please submit your revised manuscript by June 30, 2022. If you will need more time than this to complete your revisions, please reply to this message or contact the journal office at plosone@plos.org. Please include the following items when submitting your revised manuscript:A rebuttal letter that responds to each point raised by the academic editor and reviewer(s). You should upload this letter as a separate file labeled 'Response to Reviewers'.A marked-up copy of your manuscript that highlights changes made to the original version. You should upload this as a separate file labeled 'Revised Manuscript with Track Changes'.An unmarked version of your revised paper without tracked changes. You should upload this as a separate file labeled 'Manuscript'.

We look forward to receiving your revised manuscript.

Kind regards,

Wen-Wei Sung, M.D., Ph.D.

Academic Editor

PLOS ONE

Journal Requirements:

Reviewers' comments:

Reviewer's Responses to Questions

**Comments to the Author**

1. Is the manuscript technically sound, and do the data support the conclusions?

Reviewer #1: Yes

Reviewer #2: Yes

2. Has the statistical analysis been performed appropriately and rigorously? 

Reviewer #1: Yes

Reviewer #2: Yes

3. Have the authors made all data underlying the findings in their manuscript fully available?

Reviewer #1: Yes

Reviewer #2: Yes

4. Is the manuscript presented in an intelligible fashion and written in standard English?

Reviewer #1: Yes

Reviewer #2: Yes

5. Review Comments to the Author

Reviewer #1: This manuscript is well organized, and the topic has good merit to discuss. However, some issues may need to be revised before publication.

1) Minor structural or grammatical errors need to be taken into account, e.g., “Transrectal Ultrasound (TRUS), Computed Tomography (CT), and Magnetic Resonance Imaging (MRI)” in line 29 should use small letters instead of capital letters. Also, an inconsistent style has been used to present the number of patients, e.g., “Forty-eight (48) patients” in line 74 versus “22 non-recurrent patients” in line 77.

2) The citations at the end of the sentence should always be put before the full stop, e.g., Hemospermia, hematospermia, or haematospermia is defined as blood appearance in the 26 semen [1].

3) Residual confounders such as lifestyle and the influence of comedications were not considered in the current study, which should be addressed in the limitations section.

Reviewer #2: This is an interesting manuscript and providing imporatnt information. Ths following are some comments.

1. Recurrent hematospermia is an important problem in clinical practice. The authors include patients with persistent hemospermia and intractable hemospermia, with symptoms persisting over 3 months regardless of medical treatment. Is there any guideline or consensus define the definition of intractable hemospermia.

2. The succussful rate of transurethral seminal vesiculoscopy is very high in this study. Are all patients received complete TUSV? Is there any patient have receive only one side succussful TUSV

3. Intraoperative manifestations, such as Calculus, hemorrhage, mucosal lesion, were recorded. Please provide how many patient have mucosal lesion and the biopsy result.

6. PLOS authors have the option to publish the peer review history of their article (what does this mean?). If published, this will include your full peer review and any attached files.

Reviewer #1: No

Reviewer #2: No

---

## [Author Response · Author response to Decision Letter 0]

3 Jun 2022

REVIEWER 1 COMMENTS:

1. Minor structural or grammatical errors need to be taken into account, e.g., “Transrectal Ultrasound (TRUS), Computed Tomography (CT), and Magnetic Resonance Imaging (MRI)” in line 29 should use small letters instead of capital letters. Also, an inconsistent style has been used to present the number of patients, e.g., “Forty-eight (48) patients” in line 74 versus “22 non-recurrent patients” in line 77.

• Thank you for your suggestions. We have corrected our manuscript as per your comments.

2. The citations at the end of the sentence should always be put before the full stop, e.g., Hemospermia, hematospermia, or haematospermia is defined as blood appearance in the 26 semen [1].

• Thank you for your suggestion. We have corrected our manuscript as per your comments. 

3. Residual confounders such as lifestyle and the influence of comedications were not considered in the current study, which should be addressed in the limitations section.

• You have raised an important issue. We have rewritten the Limitations and future prospects section of the paper (p. 13-14, lines 192-195) to be more in line with your comments. 

REVIEWER 2 COMMENTS:

1. Recurrent hematospermia is an important problem in clinical practice. The authors include patients with persistent hemospermia and intractable hemospermia, with symptoms persisting over 3 months regardless of medical treatment. Is there any guideline or consensus define the definition of intractable hemospermia.

• You have raised an important question. We have rewritten the Introduction (p. 3, lines 32-36) to be more in line with your comments.

2. The succussful rate of transurethral seminal vesiculoscopy is very high in this study. Are all patients received complete TUSV? Is there any patient have receive only one side succussful TUSV

• You have raised two important questions. We have rewritten the Success rate section found under the Discussion heading (p. 12-13, lines 171-178) to be more in line with your comments.

3. Intraoperative manifestations, such as Calculus, hemorrhage, mucosal lesion, were recorded. Please provide how many patient have mucosal lesion and the biopsy result.

• Thank you for your suggestion. We have added the result in Table 2(p. 8) and in the Intraoperative findings section (p. 8, lines 95-98). Raw data was added in S1 File.

ETHICS STATEMENT

1. Please provide additional details regarding participant consent. In the ethics statement in the Methods and online submission information, please ensure that you have specified (1) whether consent was informed and (2) what type you obtained (for instance, written or verbal, and if verbal, how it was documented and witnessed). If your study included minors, state whether you obtained consent from parents or guardians. If the need for consent was waived by the ethics committee, please include this information.

• Thank you for your suggestion. We have rewritten the Patients section found under the Materials and Methods heading (p. 4, lines 55-56) to be more in line with your comment.

---

## [Decision Letter · Decision Letter 1]

15 Jun 2022

Intraoperative calculus or hemorrhage in transurethral seminal vesiculoscopy as a risk factor for recurrent hemospermia

PONE-D-22-12315R1

Dear Dr. Chuan-Shu Chen,

We’re pleased to inform you that your manuscript has been judged scientifically suitable for publication and will be formally accepted for publication once it meets all outstanding technical requirements.

Kind regards,

Wen-Wei Sung, M.D., Ph.D.

Academic Editor

PLOS ONE

Reviewers' comments:

Reviewer's Responses to Questions

**Comments to the Author**

1. If the authors have adequately addressed your comments raised in a previous round of review and you feel that this manuscript is now acceptable for publication, you may indicate that here to bypass the “Comments to the Author” section, enter your conflict of interest statement in the “Confidential to Editor” section, and submit your "Accept" recommendation.

Reviewer #1: All comments have been addressed

Reviewer #2: All comments have been addressed

2. Is the manuscript technically sound, and do the data support the conclusions?

Reviewer #1: (No Response)

Reviewer #2: Yes

3. Has the statistical analysis been performed appropriately and rigorously? 

Reviewer #1: (No Response)

Reviewer #2: Yes

4. Have the authors made all data underlying the findings in their manuscript fully available?

Reviewer #1: (No Response)

Reviewer #2: Yes

5. Is the manuscript presented in an intelligible fashion and written in standard English?

Reviewer #1: (No Response)

Reviewer #2: Yes

6. Review Comments to the Author

Reviewer #1: (No Response)

Reviewer #2: The authors could answer all questions. No more comments. The manuscript could be accepted and provided important information.

7. PLOS authors have the option to publish the peer review history of their article (what does this mean?). If published, this will include your full peer review and any attached files.

Reviewer #1: No

Reviewer #2: No

---

## [Editor Report · Acceptance letter]

24 Jun 2022

PONE-D-22-12315R1 

Intraoperative calculus or hemorrhage in transurethral seminal vesiculoscopy as a risk factor for recurrent hemospermia 

Dear Dr. Chen:

I'm pleased to inform you that your manuscript has been deemed suitable for publication in PLOS ONE. Congratulations! Your manuscript is now with our production department. 

Kind regards, 

on behalf of

Dr. Wen-Wei Sung 

Academic Editor

PLOS ONE